# An Optimal Hierarchical Control Strategy for 4WS-4WD Vehicles Using Nonlinear Model Predictive Control

Xuan Xu , Kang Wang, Qiongqiong Li and Jiafu Yang *

College of Mechanical and Electronic Engineering, Nanjing Forestry University, Nanjing 210037, China; xuxuan@njfu.edu.cn (X.X.)
* Correspondence: jfyang@njfu.edu.cn

**Abstract:** Advanced driving algorithms, control strategies, and their optimization in self-driving vehicles in various scenarios are hotspots in current research; 4WS-4WD (four-wheel steering and four-wheel drive) is another hotspot in the study of new concept models; and the nonlinear dynamic characteristics of self-driving vehicles (AVs) are prominent in the fast cornering mode, which leads to a significant reduction in the accuracy and stability of trajectory tracking. Based on these research backgrounds, this paper proposes a control strategy optimization idea based on the 4WS4WD vehicle and its optimization model. The main content includes the establishment of a 3D vehicle model that takes into account vehicle load transfer and position change, and the establishment of a hierarchical control strategy based on the optimized NMPC and 4WS4WD models. The controller consists of two parts: an upper tracking controller based on the new vehicle model and NMPC, and a lower decoupled controller. The tracking control effect of the algorithmic control strategy based on the model and controller is validated in the high-speed serpentine motion mode and double-shift linear motion mode on the joint simulation platform of Car Sim and Simulink.

**Keywords:** autonomous driving; 4WS-4WD; vehicle model; trajectory tracking; nonlinear model predictive control; vehicle dynamics and control



## 1. Introduction

Autonomous driving has emerged as a prominent topic for research in recent years [1–8]. Trajectory tracking is a fundamental component of autonomous driving technology that has garnered a lot of interest [9–14].

The trajectory tracking reference trajectory is contingent upon the time parameter, necessitating simultaneous control of a vehicle's longitudinal and lateral motion. This places significant demands on the control performance of trajectory tracking controllers. Scholars have undertaken numerous research investigations to produce a trajectory tracking controller that is accurate, safe, and stable for regulating autonomous driving [15–20]. Instead of considering the vehicle's own pose variation motions, such as roll and pitch, in the previously mentioned studies, researchers concentrate more on building a responsive linear vehicle model, using the vehicle's longitudinal, lateral, and transverse roll motion parameters as the controller's tracking objects. These existing studies point out that vehicles are complex systems with many interacting components, each of which has its own dynamics, and these dynamics interact with each other in complex ways. This makes the development of accurate and efficient vehicle models a difficult task. But model improvement does not mean that all relevant aspects of vehicle behavior need to be captured and all factors involved taken into account. Indeed, during high-speed driving, a vehicle's steering angle shift will result in a large variation in the vehicle's pose. This load transfer impact will further affect the vehicle's stability and increase the nonlinear response of the vehicle model [21–25]. Therefore, in this paper, the effects of load transfer and vehicle position are taken into account to improve the model so that the nonlinear characteristics of the vehicle

under high-speed turning conditions can be included in the dynamics, which is more in line with the actual working conditions.

In autonomous driving, a hierarchical control system is often employed to manage complex scenarios [26–29]. This type of system breaks down the overall task into smaller subtasks, each managed by a separate controller. The controllers are organized in a hierarchy, with higher-level controllers managing lower-level ones. This structure allows for modular design and easier debugging. In these studies, self-driving cars use hierarchical control to manage complex tasks, which can be broadly categorized into different layers of environment sensing, path planning, and trajectory tracking; there is also policy switching in different scenarios. But hierarchical controllers can be used as a way to cope with the processing of complex nonlinear models and to link the dynamic model parameter requirements with kinematic control strategies [30]. Therefore, it is necessary to consider the vehicle's pose variation and introduce it into the tracking object of the controller.

Nonlinear Model Predictive Control (NMPC) [31] is a variant of Model Predictive Control (MPC) [32] that uses nonlinear system models in prediction and has considerable applications in the unmanned driving research [33–38]. Unlike MPC, which relies on linear models, NMPC explicitly takes into account process nonlinearities and constraints, making it more suitable for a wide range of operating conditions and near the boundary of the admissible region [39,40]. Therefore, by choosing NMPC as the control algorithm and establishing the corresponding nonlinear model with constraints, the simulation can have better performance.

The novelty and contribution of this paper can be summarized as follows:

A new vehicle model based on 4WS4WD and a controller-based NMPC algorithm are used for the stability control problem of autopilot trajectory tracking. The vehicle dynamics model takes into account the 3D attitude change with load transfer; the structural parameters of the tire model are modified according to the rolling geometry of the vehicle and take into account the vertical load change of the tires due to the load transfer effect. A hierarchical control strategy is established with the upper trajectory tracking controller and the lower decoupling controller, respectively.

## 2. Nonlinear Three-Dimensional Pose-Varying Vehicle Model

The proposed nonlinear vehicle model includes a vehicle dynamics model, a chassis model, and a modified "magic formula" tire model. The vehicle dynamics model is used to analyze the state feedback of the vehicle system under the action of external forces and moments, and by selecting multiple state variables to accurately derive vehicle motion states under a given input, the chassis dynamics model is used to connect the vehicle dynamics model and the tire model, and the tire model analyzes the state feedback of the vehicle system under the action of the current state change of the vehicle system (including the change of steering wheel angle due to steering wheel angle, the change of drive tire slip rate due to longitudinal vehicle speed, etc.) and analyzes the state feedback of the external forces and moments. The tire model is used to analyze the state feedback of external forces and moments under the current state changes of the vehicle system (including the change of steering wheel angle due to steering wheel angle, the change of drive tire slip rate due to longitudinal speed, etc.), and accurately quantifies the values of external forces and moments through the model. The vehicle dynamics model, chassis model, and modified "magic formula" tire model together form a controlled closed-loop model, based on which the trajectory tracking controller provides the tire model with control quantities to achieve closed-loop control of the vehicle motion state, and the control structure is shown in Figure 1.

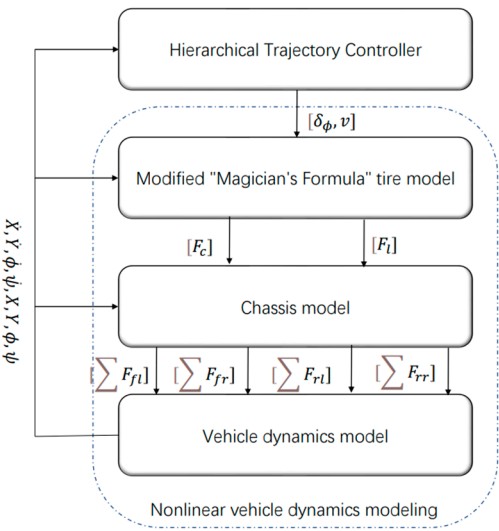

**Figure 1.** Nonlinear vehicle model and control structure block diagram.

*2.1. Model Assumptions*

The vehicle model forms the foundation of the prediction model in the NMPC controller, and the controller has demanding criteria for computational iteration speed in practical applications. Therefore, it is necessary to streamline the model to the fullest extent possible, reducing the arithmetic power required for the NMPC controller while ensuring that the prediction model possesses high accuracy.

In the dynamic modeling process, the following assumptions are required:

- Autonomous driving takes place on a flat road surface, i.e., there is no vertical freedom of the chassis due to the unevenness of the vertical road surface, and the change in the gravitational potential energy of the vehicle due to the displacement of the vertical road surface is neglected.
- The overall structure of autonomous driving is rigid, and the body stiffness is large enough.
- Assuming that the steering system is rigid and ignoring the local degrees of freedom due to the deformation of the steering column, the input from the steering wheel can be applied directly and proportionally to the steering wheel.
- The effects of drag and vehicle aerodynamics can be ignored.
- The role of tire return torque can be ignored.

*2.2. Kinematic Analysis of the Vehicle*

The research object of this paper is the overall control of the vehicle and the analysis of the vehicle's handling stability under external forces. Therefore, the vehicle body analysis can ignore vehicle reactive internal forces. Newtonian vector mechanics can be used to analyze the transmission relationship between external and internal forces to obtain the attitude of the vehicle body, which is a laborious analysis process. The analytical mechanics based on Lagrangian equations can ignore internal forces in the vehicle system by establishing generalized coordinates to analyze the work and energy conversion of the system.

Figure 2 shows a sketch of the kinematic analysis of the vehicle body. In this paper, the kinematic analysis of the vehicle body is divided into a sprung plane and an "un-sprung" plane, and four degrees of freedom are considered, including the roll degree of freedom of the sprung plane and the lateral, longitudinal, and yaw degrees of freedom of the "un-sprung" plane. The vehicle has a roll center for each of the front and rear axles, which $O_f$ is and $O_r$. The height of the front and rear axles is not equal; the vertical distance between the front axle roll center $O_f$ and the ground is $h_f$, the vertical distance between the rear axle

roll center $O_r$ and the ground is $h_r$, the connection between the roll center $O_f$ of front axle and the roll center $O_r$ of rear axle is the "roll center axis".

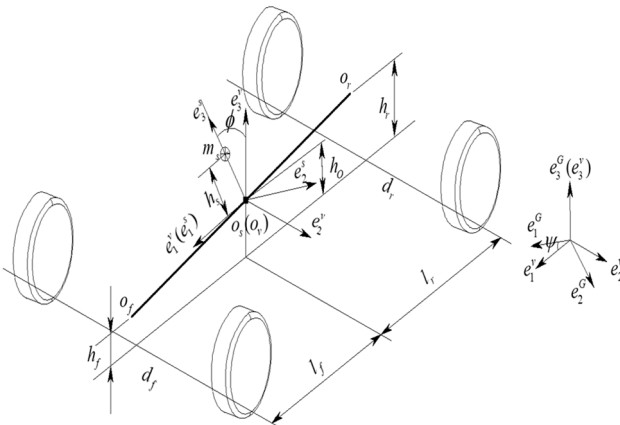

**Figure 2.** Kinematic analysis of vehicle body.

The ground reference base $e^G$, the vehicle reference base $e^v$ and the sprung plane reference base $e^s$ are established. The vector direction $e_1^v$ of the vehicle reference base $e^v$ is the direction of the vehicle forward velocity; the $e_2^v$ vector is parallel to the ground and points to the left side of the vehicle; the $e_3^v$ vector direction is perpendicular to the ground and upward, and the origin $O_v$ of the vehicle reference base $e^v$ is the intersection of the lateral plane of the center of the plane $m_s$ and the roll center axis; the lateral plane is perpendicular to the vector $e_1^v$ and the ground. The origin $O_v$ of the sprung plane reference base $e^s$ coincides with the origin $O_s$ of the vehicle reference base $e^v$.

The coordinate transformation relation between the ground reference base $e^G$, the vehicle reference base $e^v$ and the sprung plane reference base $e^s$ is

$$e^G = A^{Gv}e^v, \tag{1}$$

$$e^v = A^{vs}e^s, \tag{2}$$

where, $A^{Gv}$, $A^{vb}$ is the Directional cosine matrix.

$$A^{Gv} = \begin{pmatrix} \cos\psi & -\sin\psi & 0 \\ \sin\psi & \cos\psi & 0 \\ 0 & 0 & 1 \end{pmatrix}, \tag{3}$$

$$A^{vs} = \begin{pmatrix} 1 & 0 & 0 \\ 0 & \cos\phi & -\sin\phi \\ 0 & \sin\phi & \cos\phi \end{pmatrix}, \tag{4}$$

From Figure 1, the distance between the center of the sprung plane $m_s$ and the center axis of roll is set to $h_s$, then the position vector $P$ of the plane center $m_s$ relative to the origin $O_v$ of the vehicle reference base $e^v$ is given by

$$P = h_s \cdot e_3^s = -h_s \sin\phi \cdot e_2^v + h_s \cos\phi \cdot e_3^v, \tag{5}$$

The angle between the ground reference base $e^G$ and the vehicle reference base $e^v$ is the vehicle yaw angle $\psi$, and the angle between the sprung plane reference base $e^s$ and the vehicle reference base $e^v$ is the vehicle roll angle $\phi$, then the absolute velocity vector of the sprung plane center of plane under the inertia base $e^G$ relative to the origin $O$ of the inertia base $e^G$ is given by

$$\begin{aligned} \frac{dP^G}{dt} &= \frac{dP^s}{dt} + \omega^{Gs} \times P \\ &= \frac{dP^s}{dt} + (\omega^{Gv} + \omega^{1s}) \times P \end{aligned}, \tag{6}$$

$$\omega^{Gs} = \omega^{Gv} + \omega^{vs} = \dot{\psi} \cdot e_3^v + \dot{\phi} \cdot e_1^v, \tag{7}$$

Since the center of the sprung plane $m_s$ is at rest relative to the reference base $e^s$ of the sprung plane, $\frac{dP^s}{dt} = 0$, the absolute velocity vector is

$$\begin{aligned} \frac{dP^G}{dt} &= u_s \cdot e_1^v + v_s \cdot e_2^v + w_s \cdot e_3^v \\ &= (u_v - h_s \dot{\psi} sin\ \phi) \cdot e_1^v + \\ &\quad (v_v - h_s \dot{\phi} cos\ \phi) \cdot e_2^v \\ &\quad + (-h_s \dot{\phi} sin\ \phi) \cdot e_3^v \end{aligned}, \tag{8}$$

where $u_s, v_s, w_s$ are the forward velocity, lateral velocity, and vertical velocity of the center of the sprung plane under the inertial reference base, respectively. $u_v, v_v$ are the forward and lateral velocities of the vehicle reference base origin under the inertial reference base.

The kinetic energy of the sprung plane consists of the translational kinetic energy $E_{Ts}^t$ and the rotational kinetic energy $E_{Ts}^r$ of the sprung plane, which is

$$E_{Ts} = E_{Ts}^t + E_{Ts}^r = \frac{1}{2} m_s \left\| \frac{dP^G}{dt} \right\|_2 + \frac{1}{2} \left( \omega^{Gs} \right)^T I_s \omega^{Gs} P, \tag{9}$$

where $I_s$ is the rotational inertia matrix of the sprung plane, and since the vehicle is symmetric about its centerline, $I_s$ is

$$I_s = \begin{pmatrix} I_{xx} & 0 & -I_{xz} \\ 0 & I_{yy} & 0 \\ -I_{xz} & 0 & I_{zz} \end{pmatrix}, \tag{10}$$

where $I_{ii}$ is the rotational inertia of the sprung plane around the vector $e_i^s$, and $I_{ij}$ is the product of inertia. According to the reference base transformation matrix, $\omega^{Gs}$ in the reference base on the sprung $e^s$ is

$$\omega^{Gs} = \dot{\phi} \cdot e_1^s + \dot{\psi} sin\ \phi \cdot e_2^s + \dot{\psi} cos\ \phi \cdot e_3^s, \tag{11}$$

Take Equations (8), (10), and (11) into account to obtain the kinetic energy of the sprung plane, $E_{TS}$ and the kinetic energy of translation and rotation, respectively.

$$\begin{aligned} E_{Ts}^t &= \frac{1}{2} m_s \left\| \begin{pmatrix} u_v + h_s \dot{\psi} \sin \phi \\ v_v - h_s \dot{\phi} \cos \phi \\ -h_s \dot{\phi} \sin \phi \end{pmatrix} \right\|_2 \\ &= \frac{1}{2} m_s [(u_v^2 + h_s^2 \dot{\psi}^2 \sin^2 \phi + 2 u_v h_s \dot{\psi} \sin \phi) + \\ &\quad (v_v^2 + h_s^2 \dot{\phi}^2 \cos^2 \phi - 2 v_v h_s \dot{\phi} \cos \phi) + (h_s^2 \dot{\phi}^2 \sin^2 \phi)] \\ &= \frac{1}{2} m_s [(u_v^2 + v_v^2) + (h_s^2 \dot{\psi}^2 \sin^2 \phi + h_s^2 \dot{\phi}^2 \sin^2 \phi) - \\ &\quad (2 v_v h_s \dot{\phi} \cos \phi - 2 u_v h_s \dot{\psi} \sin \phi + h_s^2 \dot{\phi}^2 \cos^2 \phi)] \end{aligned}, \tag{12}$$

In (12), when the second term of the equation $\phi$ is small, $\sin \phi \approx \phi$, then the second term of the equation above, which is a fourth-order term, can be ignored.

$$\begin{aligned} E_{Is}^r &= \frac{1}{2} \begin{pmatrix} \dot{\phi} \\ \dot{\psi} \sin \phi \\ \dot{\psi} \cos \phi \end{pmatrix}^T \times \begin{pmatrix} I_{xx} & 0 & -I_{xz} \\ 0 & I_{yy} & 0 \\ -I_{xz} & 0 & I_{zz} \end{pmatrix} \times \begin{pmatrix} \dot{\phi} \\ \dot{\psi} \sin \phi \\ \dot{\psi} \cos \phi \end{pmatrix} \\ &= \frac{1}{2} \left( I_{xx} \dot{\phi}^2 - 2 I_{xz} \dot{\phi} \dot{\psi} \cos \phi + I_{yz} \dot{\psi}^2 \sin^2 \phi + I_{zz} \dot{\psi}^2 \cos^2 \phi \right) \end{aligned}, \tag{13}$$

In the equation above, the small second and third terms are ignored, and the first term of the roll moment of inertia and the fourth term of the yaw moment of inertia are mainly considered.

The under-sprung plane of a vehicle is almost entirely composed of two chassis components: the front suspension and front axle, and the rear suspension and rear axle. The distance between the front and rear chassis components is far apart, and in the case of a large angular velocity of the vehicle roll, treating the front and rear chassis components as one rigid body will cause large calculation errors. Therefore, the under-sprung plane is divided into front under-sprung plane $m_f$ and rear under-sprung plane $m_r$, and the under-sprung plane kinetic energy is divided into front under-sprung plane kinetic energy $E_{Tf}$ and rear under-sprung plane kinetic energy $E_{Tr}$, and each part of kinetic energy includes both translational kinetic energy and rotational kinetic energy.

$$E_{If} = \frac{1}{2}m_f\left(u_f^2 + v_f^2\right) + \frac{1}{2}I_{z=f}\dot{\psi}^2, \tag{14}$$

$$E_{T_r} = \frac{1}{2}m_r\left(u_r^2 + v_r^2\right) + \frac{1}{2}I_{zzr}\dot{\psi}^2, \tag{15}$$

where $u_f = u_v, u_r = u_v, v_r = v_{v-}l_r\dot{\psi}$ and $v_f = v_v + l_f\dot{\psi}$.

Assuming that the roll stiffness coefficient $K_\phi$ and the roll damping coefficient $C_\phi$ of the suspension are constant when the body is rolled, the dissipation energy of the vehicle is

$$E_D = \frac{1}{2}C_\phi\dot{\phi}^2, \tag{16}$$

The potential energy of the vehicle $E_V$ includes two parts: one is the front and rear suspension elastic potential energy generated by the body roll, and the other is the gravitational potential energy of the sprung plane when the body rolls. The change of the body roll angle directly affects the elastic potential energy, while the change of the roll angle will cause the height of the sprung plane center to change, thus affecting the gravitational potential energy. Therefore, the gravitational potential energy is

$$\begin{aligned} E_V &= E_{Vk} + E_{Tg} \\ &= \frac{1}{2}K_\phi\phi^2 - m_sgh_s(1 - \cos\phi) \end{aligned}, \tag{17}$$

where $E_{Vk}$ is the elastic potential energy, $E_{Vg}$ is the gravitational potential energy, and $g$ is the gravitational acceleration.

### 2.3. Vehicle Dynamics Model

The Lagrangian equation with dissipative terms is

$$\frac{d}{dt}\left(\frac{\partial E_T}{\partial q}\right) - \frac{\partial E_T}{\partial q} + \frac{\partial E_V}{\partial q} + \frac{\partial E_D}{\partial q} = F_q, \tag{18}$$

where, $q = (xy\psi\phi)^T$ is the generalized coordinate vector, where $x$, $y$ are the longitudinal and lateral coordinates under the vehicle reference base, $\dot{x} = u_v$, $\dot{y} = v_v$, and are the generalized force vectors $F_q = \left(F_x\ F_y\ M_\psi\ M_\phi\right)^T$.

The generalized force is analyzed, and the longitudinal generalized force $F_x$ under the vehicle reference base is

$$F_x = F_{xf} + F_{xr}, \tag{19}$$

where $F_{xf}$ is the combined force of the longitudinal force components provided by the two tires on the front axle, and $F_{xr}$ is the combined force of the longitudinal force components provided by the two tires on the rear axle.

Similarly, the lateral generalized force $F_y$ under the vehicle reference base is

$$F_y = F_{yf} + F_{yr}, \tag{20}$$

where $F_{yf}$ is the combined force of the lateral force components provided by the two tires on the front axle, and $F_{yr}$ is the combined force of the lateral force components provided by the two tires on the rear axle.

The vehicle yaw generalized moment, $M_\psi$ generated by the lateral force of the front and rear tires, is given by

$$M_\psi = l_f F_{yf} - l_r F_{yr}, \tag{21}$$

where $l_f$ is the longitudinal distance from the center of the plane to the front axle, and $l_r$ is the longitudinal distance from the center of the plane to the rear axle.

The vehicle roll generalized moment $M_\phi$ generated by the lateral force of the front and rear tires is given by

$$M_\phi = \left(h_O - h_f\right)F_{yf} + (h_O - h_r)F_{yr}, \tag{22}$$

where, $h_f$ is the vertical distance between the roll center of the front axis and the ground, $h_r$ is the vertical distance between the roll center of the rear axis and the ground, and $h_O$ is the vertical distance between the origin and the ground.

The kinetic energy, potential energy, and dissipative energy are applied to the Lagrangian equation and to the generalized coordinate vector $q$ and its first-order derivative to find the partial derivative. The Lagrangian model of the vehicle is obtained as

Vehicle longitudinal (along $x$ axis) equation of motion

$$\left(m_f + m_s + m_r\right)\left(\ddot{x} - \dot{y}\dot{\psi}\right) - m_s h_s\left(\ddot{\psi}sin\,\phi + \dot{\psi}\dot{\phi}cos\,\phi\right) = F_{xf} + F_{xr}, \tag{23}$$

Vehicle lateral motion (along $y$ the axis) equation

$$\left(m_f + m_s + m_r\right)\left(\ddot{y} + \dot{x}\dot{\psi}\right) + m_s h_s\left(\dot{\phi}^2 cos\,\phi - \ddot{\phi}cos\,\phi\right) + \left(m_f l_f - m_r l_r\right)\ddot{\psi} = F_{yf} + F_{yr}, \tag{24}$$

Vehicle yaw motion (around $z$ the axis) equation

$$\begin{aligned}\left(I_{zz} + I_{zzf} + I_{zzr} + m_f l_f^2 + m_r l_r^2\right)\ddot{\psi} + \left(m_f l_f - m_r l_f\right)\left(\ddot{y} + \dot{x}\dot{\psi}\right) \\ -m_s h_s\left(\ddot{x}sin\,\phi + \dot{x}\dot{\phi}cos\,\phi\right) + I_{xz}\ddot{\phi} = l_f F_{yf} - l_r F_{yr},\end{aligned} \tag{25}$$

Vehicle roll motion (around the $x$ axis) equation

$$\begin{aligned}m_s h_s\left(\ddot{y} + \dot{x}\dot{\psi}\right) + \left(I_{xx} + m_s h_s^2\right)\ddot{\phi} - m_s h_s\left(\dot{x}\dot{\psi}cos\,\phi - \dot{y}\dot{\phi}sin\,\phi - h_s\dot{\phi}^2 cos\,\phi sin\,\phi\right) \\ +2I_{zz}\dot{\psi}^2 cos\,\phi sin\,\phi + \left(K_\phi\phi - m_s h_s g sin\,\phi\right) + C_\phi\dot{\phi} + I_{xz}\ddot{\psi} = \left(h_O - h_f\right)F_{yf} + (h_O - h_r)F_{yr},\end{aligned} \tag{26}$$

For this Lagrangian model, an appropriate simplification can be carried out by ignoring the higher-order terms when $\phi$ is small and considering $\sin\phi \approx \phi$, $\cos\phi \approx 1$ then, the simplified model is obtained as.

Vehicle longitudinal (along the $x$ axis) equation of motion

$$\left(m_f + m_s + m_r\right)\left(\ddot{x} - \dot{y}\dot{\psi}\right) = F_{xf} + F_{xr}, \tag{27}$$

Vehicle lateral motion (along $y$ the axis) equation

$$\left(m_f + m_s + m_r\right)\left(\ddot{y} + \dot{x}\dot{\psi}\right) - m_s h_s\ddot{\phi} + \left(m_f l_f - m_r l_r\right)\ddot{\psi} = F_{yf} + F_{yr} \tag{28}$$

Vehicle yaw motion (around $x$ the axis) equation

$$\left(I_{zz} + I_{zzf} + I_{zzr} + m_f l_f^2 + m_r l_r^2\right)\ddot{\psi} + \left(m_f l_f - m_r l_r\right)\left(\ddot{y} + \dot{x}\dot{\psi}\right) - I_{xz}\ddot{\phi} = l_f F_{yf} - l_r F_{yr}, \quad (29)$$

Vehicle roll motion (around the $x$ axis) equation

$$m_s h_s\left(\ddot{y} + \dot{x}\dot{\psi}\right) + \left(I_{xx} + m_s h_s^2\right)\ddot{\phi} + (K_\phi - m_s h_s g)\phi + C_\phi\dot{\phi} + I_{xz}\ddot{\psi} = \left(h_O - h_f\right)F_{yf} + (h_O - h_r)F_{yr}. \quad (30)$$

*2.4. Chassis Dynamics Model*

The chassis dynamics model is used to connect the vehicle body dynamics with the tire model, decouple the coupled state variables of the vehicle body and input them into each tire model, and integrate the longitudinal and lateral forces output from each tire model into the combined forces applied to the vehicle body. The chassis dynamics model is shown in Figure 3. The tire forces in each independent tire reference system can be translated into the vehicle reference system and ground reference system, and the tire forces and velocities of each tire are expressed as: $z_{\Delta,\Omega S}$. The $z_{\Delta,\Omega S}$ is the state variables, $Z \in \{F, v, \delta\}$, $F$ is the tire force, $v$ is the tire velocity, and $\delta$ is the wheel angle; $\Delta$ is the reference direction, $\Delta = \{l, c, w, y, z, X, Y\}$, $l$ and $c$ are the longitudinal and lateral directions of the tire reference system, $x$ and $y$ are the $x$ and $y$ axis directions of the vehicle reference system, $z$ is the $z$ axis direction of the vehicle reference system, $X$ and $Y$ are the $X$ and $Y$ axis directions of the ground reference system; $\Omega$ is the axle positioning, $\Omega = \{f, r\}$, $f$ and $r$ are the front and rear axles; $S$ is the side positioning, $S = \{l, r\}$, $l$ $l$ and $r$ are the left side of the body and the right side of the body.

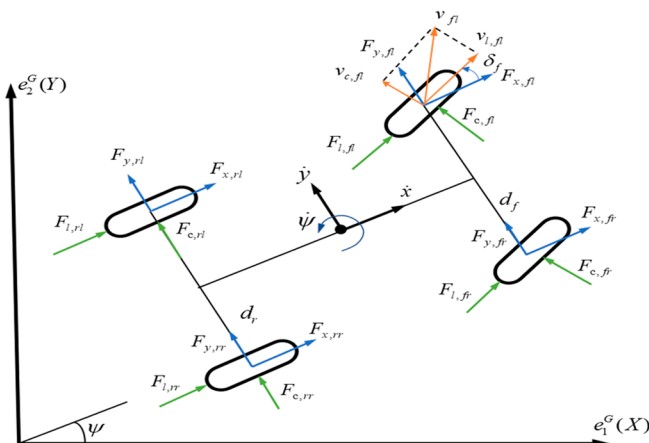

**Figure 3.** Chassis dynamics model.

The transformation relation between the longitudinal and lateral forces of the tire in the vehicle reference system and the ground reference system is

$$F_{x,\Omega S} = F_{l,\Omega S}\cos\delta_{\Omega S} - F_{c,\Omega S}\sin\delta_{\Omega S}, \quad (31)$$

$$F_{y,\Omega S} = F_{c,\Omega S}\cos\delta_{\Omega S} + F_{l,\Omega S}\sin\delta_{\Omega S}, \quad (32)$$

$$F_{X,\Omega S} = F_{l,\Omega S}(\cos\delta_{\Omega S}\cos\psi - \cos\delta_{\Omega S}\sin\psi) - F_{c,\Omega S}(\sin\delta_{\Omega S}\cos\psi + \cos\delta_{\Omega S}\sin\psi), \quad (33)$$

$$F_{Y,\Omega S} = F_{l,\Omega S}(\cos\delta_{\Omega S}\cos\psi - \cos\delta_{\Omega S}\sin\psi) - F_{c,\Omega S}(\sin\delta_{\Omega S}\cos\psi + \cos\delta_{\Omega S}\sin\psi), \quad (34)$$

The longitudinal velocity of the tire is transformed in the vehicle reference system through the following relationship:

$$v_{l,fl} = \left(\dot{x} - d_f\dot{\psi}\right)\cos\delta_f + \left(\dot{y} + l_f\dot{\psi}\right)\sin\delta_f v_{l,fr} = \left(\dot{x} + d_f\dot{\psi}\right)\cos\delta_f + \left(\dot{y} + l_f\dot{\psi}\right)\sin\delta_f v_{l,rl}$$
$$= \left(\dot{x} - d_r\dot{\psi}\right)\cos\delta_r + \left(\dot{y} - l_r\dot{\psi}\right)\sin\delta_r v_{l,rr} = \left(\dot{x} + d_r\dot{\psi}\right)\cos\delta_r + \left(\dot{y} - l_r\dot{\psi}\right)\sin\delta_r \quad (35)$$

where $d_f$ is the front half wheelbase, $d_r$ is the rear half wheelbase; $l_f$ is the distance from the center of plane to the front axle, $l_r$ is the distance from the center of plane to the rear axle.

The expressions for the vertical load of the tire are given as

$$F_{z,fl} = \frac{ml_r g - m(h_s + h_O)\ddot{x}}{2\left(l_f + l_r\right)} - \frac{ml_r(h_s + h_O)\ddot{y}}{2d_f\left(l_f + l_r\right)}, \tag{36}$$

$$F_{z,fr} = \frac{ml_r g - m(h_s + h_O)\ddot{x}}{2\left(l_f + l_r\right)} + \frac{ml_r(h_s + h_O)\ddot{y}}{2d_f\left(l_f + l_r\right)}, \tag{37}$$

$$F_{z,rl} = \frac{ml_f g + m(h_s + h_O)\ddot{x}}{2\left(l_f + l_r\right)} - \frac{ml_f(h_s + h_O)\ddot{y}}{2d_r\left(l_f + l_r\right)}, \tag{38}$$

$$F_{z,rr} = \frac{ml_f g + m(h_s + h_O)\ddot{x}}{2\left(l_f + l_r\right)} + \frac{ml_f(h_s + h_O)\ddot{y}}{2d_r\left(l_f + l_r\right)}, \tag{39}$$

where the calculation of the vertical load takes into account the lateral and longitudinal acceleration.

### 2.5. Modified "Magic Formula" Tire Model

The commonly used tire models include both empirical and physical models, and the commonly used empirical models include the power exponential unified tire model and the "magic formula" tire model [41], while the commonly used physical models include the "string model" and the "brush model". Among them, the "magic formula" model based on the function fitting method uses a form to express both longitudinal and lateral tire forces, which is convenient for subsequent control algorithm design and program writing. The "magic formula" tire model has been continuously studied and used by researchers [42–46]. Before we can improve it, we need to introduce the variables we need to use and their definitions. The general expressions of the "magic formula" tire model are:

$$y = Dsin\{Carctan[Bx - E(Bx - arctanBx)]\} + S_v, \tag{40}$$

To calculate the longitudinal force and lateral force, the equations are:

(1) Longitudinal force calculation: $y$ is considered as the longitudinal tire force $F_l$; $x$ is the sum of the tire slip rate $s$ and horizontal drift factor $S_h$; $C$ is the curve shape factor; $B$ is the stiffness factor; $B$ is the peak factor; $E$ is the curvature factor; $S_v$ is the vertical drift factor.

The expression of each coefficient is given by

$$\begin{cases} C = L_0 \\ D = L_1 F_z^2 + L_2 F_z \\ B = \left(L_3 F_z^2 + L_4 F_z\right) \times e^{-L_5 F_z} / (C \times D) \\ E = L_6 F_z^2 + L_7 F_z + L_8 \\ S_h = L_9 F_z + L_{10} \\ S_v = 0 \end{cases}, \tag{41}$$

where the longitudinal force characteristics parameters include $L_0 \sim L_{10}$ and the variables include tire vertical force, $F_z$, tire slip rate $s$.

The slip rate $s$ is defined as

$$\begin{cases} drive : s = \frac{r_w \omega_w - v_l}{r_w \omega_w} \\ braking : s = \frac{v_l - r_w \omega_w}{v_l} \end{cases}, \tag{42}$$

where $r_w$ is the radius of the tire, $\omega_w$ is the angular velocity of the wheel, $v_l$ is the longitudinal velocity of the wheel.

From Equations (33)–(40), both the tire slip rate and the longitudinal force of the tire can be expressed by the function

$$s = s\left(\dot{x}, \dot{y}, \dot{\psi}, \delta, \omega_w\right), \tag{43}$$

$$F_l = f_l(F_z, s), \tag{44}$$

(2) Lateral force calculation:

The letter $y$ is denoted for the tire lateral force $F_c$; $x$ is the sum of the tire lateral deflection angle $\alpha$ and horizontal drift factor $S_h$; $C$ is the curve shape factor; $B$ is the stiffness factor; $D$ is the peak factor; $E$ is the curvature factor; and $S_v$ is the vertical drift factor.

The expressions of each coefficient are given as

$$\begin{cases} C = P_0 \\ D = P_1 F_z^2 + P_2 F_z \\ B = P_3 \sin\left(2 \arctan\frac{F_z}{P_4}\right) \times (1 - P_5|\gamma|)/(C \times D) \\ E = P_6 F_z + P_7 \\ S_h = P_9 F_z + P_{10} + P_8 \gamma \\ S_v = P_{11} F_z \gamma + P_{12} F_z + P_{13} \end{cases}, \tag{45}$$

where, the front tire slip angle $\alpha_f$ is

$$\alpha_f = a_1\left(\dot{x}, \dot{y}, \dot{\psi}, \delta_f\right) = \arctan\left(\frac{\dot{y} + l_f \dot{\psi}}{\dot{x}}\right) - \delta_f, \tag{46}$$

The rear tire slip angle $\alpha_r$ is

$$\alpha_r = a_2\left(\dot{x}, \dot{y}, \dot{\psi}, \delta_r\right) = \arctan\left(\frac{\dot{y} - l_r \dot{\psi}}{\dot{x}}\right) - \delta_r, \tag{47}$$

From Equation (33)~(40), the tire lateral force function is

$$F_c = f_c(F_z, \alpha_i, \gamma), (i = f, r), \tag{48}$$

(3) Correction of alignment change: Pose variation will lead to changes in wheel positioning parameters, further causing changes in the wheel angle and camber of the wheels, which will have an impact on the tire force and its balance.

These two deformations are collectively referred to as alignment changes. Therefore, it is necessary to modify the tire model so that it can accurately reflect the actual working state of the tire under extreme conditions such as steering at high speed and sideslips with a large roll angle.

(1) Roll steering

Roll steering caused by a roll during a change of attitude depends on the specific structure of the suspension. Roll-steering of a rigid axle suspension is called axle steering, i.e., the axle turns while the tire does not actually turn. However, from a kinematic point of view, the effect of axle rotation is equivalent to tire rotation. The roll steering of an independent suspension can be obtained by analyzing the geometry of the mechanism.

A common roll-steering curve is shown in Figure 4. A roll steering condition that tends to steer to the inside of the vehicle is called toe-in, and a condition that tends to steer to the outside is called toe-out. The roll steering is equivalent to superimposing the wheel angle $\delta_\phi$ on all four wheels, as shown in Figure 4. The wheel angle $\delta_\phi$ is related to the distance $d_s$ of suspension moving, which can be obtained from the roll angle $\phi$ and the suspension geometry.

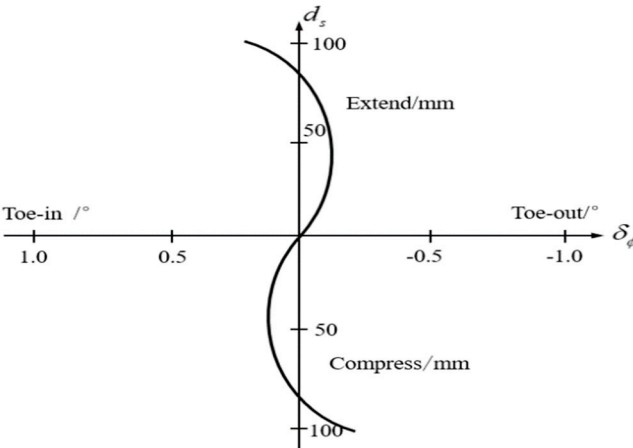

**Figure 4.** Roll steering curve.

The expression for the wheel angle $\delta_\phi$ is given by

$$\delta_\phi = \frac{\partial \delta}{\partial d_s} \cdot \frac{\partial d_s}{\partial \phi} \phi, \tag{49}$$

As seen in Figure 4, when the motion travel $d_s$ of the suspension is small, i.e., the roll angle $\phi$ is not large, the wheel angle $\delta_\phi$ and the roll angle $\phi$ can be considered to be nearly linear, so the (49) can be approximated as

$$\delta_\phi = K_\delta \phi, \tag{50}$$

where $K_\delta$ is the roll steering stiffness.

The front axle suspension and the rear axle suspension have different roll steering stiffness coefficients due to their different geometries. According to the reference base, the left suspension stretches and the right suspension compresses when the roll angle $\phi$ is positive; thus, the left wheel roll steering angle $\delta_{\phi,l}$ is the toe-in and the right wheel roll steering angle $\delta_{\phi,R}$ is the toe-out.

Extending (50) to the four-wheel case leads to

$$\begin{cases} \delta_{\phi,fl} = -K_{\delta,f}\phi, \\ \delta_{\phi,fr} = K_{\delta,f}\phi \\ \delta_{\phi,rl} = -K_{\delta,r}\phi \\ \delta_{\phi,rr} = K_{\delta,r}\phi \end{cases}, \tag{51}$$

The effect of roll steering on the longitudinal force of the wheel is small, but it has a large effect on the lateral force. The corrected wheel angle and tire slip angle are

$$\delta^* = \delta + \delta_\phi, \tag{52}$$

$$\alpha^* = a\left(\dot{x}, \dot{y}, \dot{\psi}, \delta^*\right), \tag{53}$$

(2) Camber change

When the suspension is a rigid axle suspension, the pose variation does not cause the wheels to change camber. When the suspension is an independent suspension, the camber change causes the wheels to change camber. The camber change curve of an independent suspension can be obtained by a geometric analysis of the guiding mechanism. The camber change curve of the double wishbone suspension is shown in Figure 5, and the camber change is equivalent to adding a camber change angle $\gamma_\phi$ on the wheels.

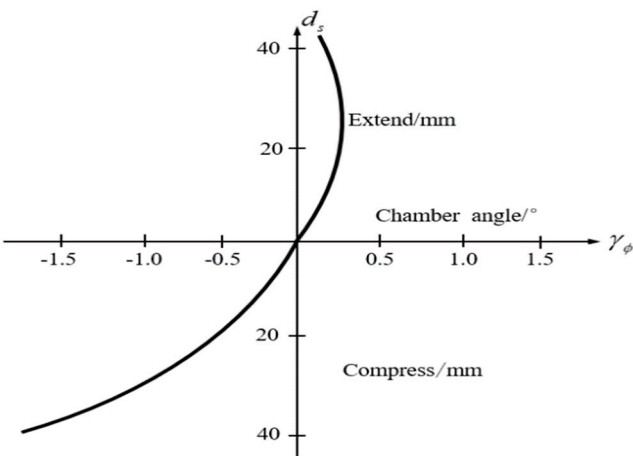

**Figure 5.** Camber change curve.

The expression for the angle of change of camber $\gamma_\phi$ is

$$\gamma_\phi = \frac{\partial \gamma}{\partial d_s} \cdot \frac{\partial d_s}{\partial \phi} \phi, \tag{54}$$

As seen in Figure 5, when the motion travel of $d_s$ the suspension is small, i.e., the roll angle $\phi$ is not large, the change of camber angle $\gamma_\phi$ can be considered to be nearly linear with the roll angle $\phi$, and (54) can be approximated by

$$\gamma_\phi = K_\gamma \phi, \tag{55}$$

where $K_\gamma$ is the camber change stiffness. Thus, (55) for the four-wheels case is expanded as

$$\begin{cases} \gamma_{\phi,fl} = K_{\gamma,f}\phi, \\ \gamma_{\phi,fr} = -K_{\gamma,f}\phi \\ \gamma_{\phi,rl} = K_{\gamma,r}\phi \\ \gamma_{\phi,rr} = -K_{\gamma,r}\phi \end{cases}, \tag{56}$$

In the case of no roll, the camber angle of the tire is small, so the initial camber angle $\gamma$ is set to 0. Therefore, the corrected camber angle $\gamma^*$ is expressed as

$$\gamma^* = \gamma_\phi = K_\gamma \phi, \tag{57}$$

(3) Modification of tire vertical force:

The variation of the vehicle's spring mass with the change in vehicle attitude is shown in Figure 6. Since the center of the sprung plane $m_s$ is not on the roll center axis but above it, the distance between them is $h_s$, thus when the car body is rolled, the position of the center of the plane $m_s$ will be laterally offset, resulting in a change in the vertical load of the left and right wheels. The lateral offset of the center of plane $m_s$ is $\Delta d$, the distance between the center of plane $m_s$ and the roll center $O_s$ is $h_s$.

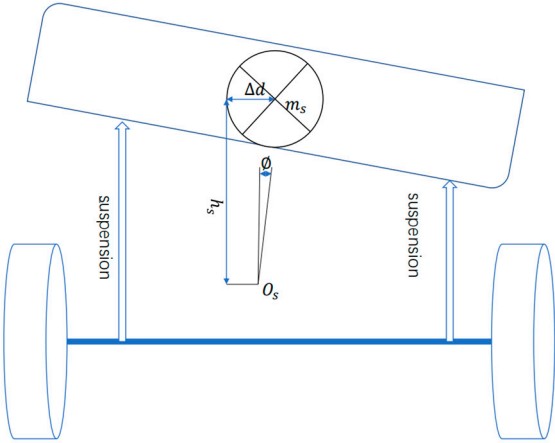

**Figure 6.** Rear view of car body roll.

The expression for the lateral offset $\Delta d$ is

$$\Delta d = h_s sin\ \phi, \tag{58}$$

The vehicle roll will be in the left and right wheel additional tire vertical force $\Delta F_z$, can be obtained from the rolling coefficient $\lambda_z$ and the original tire vertical load $F_z$, the expression is

$$\Delta F_Z = \lambda_Z F_Z, \tag{59}$$

The factor $\lambda_z$ is determined by the lateral offset $\Delta d$ and the wheel track $d$, and the expression is

$$\lambda_{z,i} = (-1)^j \frac{2\Delta d}{d}, \tag{60}$$

where $i$ is $l$ for the left wheel and $i$ is $r$ for the right wheel; when $i = l$, $j = 1$; when $i = r$, $j = 0$.

In summary, the modified vertical load is obtained as

$$F_z^* = F_z + \Delta F_z = F_z + (-1)^j \frac{2\Delta d}{d} F_z, \tag{61}$$

This chapter establishes the chassis dynamics model of the vehicle based on the on-spring mass reference base, the under-spring mass reference base, and the ground inertia reference base established by the vehicle body, which is used to decouple the coupled state quantities of the vehicle body and input them into each tire model. At the same time, the longitudinal and lateral forces outputted from each tire model are integrated into the combined force exerted on the vehicle body. Through the method of internal force analysis of the structure and the method of kinematics analysis of the vehicle body, the mechanical model of the tire's drooping load is deduced and considered. The influence of the load transfer effect on the vertical load of tires was thus corrected. The purpose is to sort out the relationship and mutual influence of each part of the model.

## 3. Upper Trajectory Tracking Controller

The upper trajectory tracking control based NMPC includes four parts: prediction model, constraints, cost function, and solver.

### 3.1. Predictive Model

The prediction model based on the vehicle model in this paper is strongly nonlinear. Moreover, the vehicle dynamics include the nonlinear characteristics of the pose variation; the chassis model has the nonlinear coupling characteristics of the chassis; and the modified "magic formula" tire model has the nonlinear longitudinal and lateral force output char-

acteristics of the tire. The three vehicle models are dynamically coupled with each other, which accurately reflects the pose variation and dynamic characteristics of the vehicle and improves the performance of the controller.

Combining Equations (27)–(34), (46), (50), and (54), the equations of the nonlinear vehicle model can be obtained, and after solving the derivative terms of the state variables, the state space equations of the nonlinear prediction model are obtained as follows:

$$
\begin{cases} \dot{\xi} = f(\xi, u) \\ \eta = g(\xi) \end{cases},
\tag{62}
$$

where $\xi = \left( \dot{x}, \dot{y}, \dot{\phi}, \dot{\psi}, X, Y, \phi, \psi \right)$ is the eight-dimensional state variables; $\eta = \left( \dot{X}, \dot{Y}, \dot{\phi}, \dot{\psi}, X, Y, \phi, \psi \right)$ is the eight-dimensional output variables; $u = (\delta, a_x)$ is the equivalent two-dimensional control variables; $\delta$ is equivalent wheel deflection angle; and $a_x$ is equivalent longitudinal acceleration. In the designed dynamics model, the second-order derivative terms and the first-order derivative terms are coupled with each other, i.e., the derivatives of the state quantities are not decoupled from the state quantities, and it is not possible to construct the prediction model of the NMPC controller directly. Therefore, computational decoupling is used to separate the derivative terms of the state quantities to obtain a decoupled nonlinear prediction model.

Decoupling state space equations as follows:

$$
\ddot{x} = \frac{F_{xf} + F_{xr} + M\dot{\psi}\dot{y}}{M},
\tag{63}
$$

$$
\ddot{y} = \begin{pmatrix} C_\phi I_{xz} M\phi - C_\phi IM\phi + C_\phi\phi l_f^2 m_f^2 \\ + C_\phi\phi l_r^2 m_r^2 + F_{yf} IMh_f \\ F_{yf} IMh_o - F_{yr} IMh_o + F_{yr} IMh_r - \\ F_{yf} I_{xz} Mh_f + F_{yf} I_{xz} Mh_o + \\ F_{yr} I_{xz} Mh_o - F_{yr} I_{xz} Mh_r - \\ F_{yf} h_f l_f^2 m_f^2 + F_{yf} h_o l_f^2 m_f^2 + \\ F_{yr} h_o l_f^2 m_f^2 - F_{yr} h_r l_f^2 m_f^2 - \\ F_{yf} h_f l_r^2 m_r^2 + F_{yf} h_o l_r^2 m_r^2 + F_{yr} h_o l_r^2 m_r^2 \\ - F_{yr} h_r l_r^2 m_r^2 + F_{yf} I_{xz} M l_f - \\ F_{yr} I_{xz} M l_r + I K_\phi M\phi - I_{xz} K_\phi M\phi - \\ K_\phi l_f^2 m_f^2 \phi - K_\phi l_r^2 m_r^2 \phi + F_{yf} I h_s m_s \\ + F_{yr} I h_s m_s - F_{yf} I_{xz} h_s m_s - \\ F_{yr} I_{xz} h_s m_s - F_{yf} I_{xz} l_f m_f - F_{yr} I_{xz} l_f m_f \\ + F_{yf} I_{xz} l_r m_r + F_{yr} I_{xz} l_r m_r - \\ F_{yf} h_s l_f^2 m_f m_s - F_{yr} h_s l_r^2 m_r m_s - \\ IMgh_s m_s \phi + I_{xz} Mgh_s m_s \phi + \\ gh_s l_f^2 m_f^2 m_s \phi + gh_s l_r^2 m_r^2 m_s \phi - \\ 2C_\phi\phi l_f l_r m_f m_r + 2F_{yf} h_f l_f l_r m_f m_r - \\ 2F_{yf} h_o l_f l_r m_f m_r - 2F_{yr} h_o l_f l_r m_f m_r + \\ 2F_{yr} h_r l_f l_r m_f m_r + F_{yr} h_s l_f l_r m_f m_s + \\ F_{yf} h_s l_f l_r m_f m_s + 2K_\phi l_f l_r m_f m_r \phi \\ - 2gh_s l_f l_r m_f m_r m_s \phi \end{pmatrix} \Bigg/ \begin{pmatrix} I_{xZ} h_s^2 m_s^2 - Ih_s^2 m_s^2 - II_{xX} M \\ + I_{xx} I_{xZ} M + I_{xx} l_f^2 m_f^2 + \\ I_{xx} l_r^2 m_r^2 - IMh_s^2 m_s + \\ I_{xZ} Mh_s^2 m_s + h_s^2 l_f^2 m_f^2 m_s + \\ h_s^2 l_r^2 m_r^2 m_s + I_{Xz} h_s l_f m_f m_s \\ - I_{Xz} h_s l_r m_r m_s - \\ 2I_{xx} l_f l_r m_f m_r - 2h_s^2 l_f l_r m_f m_r m_s \end{pmatrix}
\tag{64}
$$

$$\ddot{\psi} = \begin{pmatrix} F_{yf}h_s^2l_fm_s^2 - F_{yr}h_s^2l_rm_s^2 + F_{yf}I_{xx}Ml_f \\ -F_{yr}I_{xx}Ml_r - F_{yf}I_{xx}l_fm_f- \\ F_{yr}I_{xx}l_fm_f + F_{yf}I_{xx}l_rm_r + F_{yr}I_{xx}l_rm_r \\ +F_{yf}Mh_s^2l_fm_s - F_{yr}Mh_s^2l_rm_s- \\ F_{yf}h_s^2l_fm_fm_s - F_{yr}h_s^2l_fm_fm_s+ \\ F_{yf}h_s^2l_rm_rm_s + F_{yr}h_s^2l_rm_rm_s- \\ gh_s^2l_fm_fm_s^2\phi + gh_s^2l_rm_rm_s^2\phi- \\ C_\phi\dot\phi h_sl_fm_fm_s + C_\phi\dot\phi h_sl_rm_rm_s+ \\ F_{yf}h_fh_sl_fm_fm_s - F_{yf}h_oh_sl_fm_fm_s \\ -F_{yr}h_oh_sl_fm_fm_s + F_{yr}h_rh_sl_fm_fm_s- \\ F_{yf}h_fh_sl_rm_rm_s + F_{yf}h_oh_sl_rm_rm_s \\ +F_{yr}h_oh_sl_rm_rm_s- \\ F_{yr}h_rh_sl_rm_rm_s+ \\ K_\phi h_sl_fm_fm_s\phi - K_\phi h_sl_rm_rm_s\phi \end{pmatrix} \bigg/ \begin{pmatrix} I_{xz}h_s^2m_s^2 - Ih_s^2m_s^2 - II_{xx}M \\ +I_{xx}I_{xz}M + I_{xx}l_f^2m_f^2+ \\ I_{xx}l_r^2m_r^2 - IMh_s^2m_s + I_{xz}Mh_s^2m_s \\ +h_s^2l_f^2m_f^2m_s + h_s^2l_r^2m_r^2m_s+ \\ I_{xz}h_sl_fm_fm_s - I_{xz}h_sl_rm_rm_s \\ -2I_{xx}l_fl_rm_fm_r - 2h_s^2l_fl_rm_fm_rm_s \end{pmatrix} \tag{65}$$

$$\dot{X} = \dot{x}cos\psi - \dot{y}sin\psi, \tag{66}$$

$$\dot{Y} = \dot{x}sin\psi - \dot{y}cos\psi, \tag{67}$$

The above prediction model has been decoupled with the first-order derivative terms of the state quantities on the left-hand side of the equation, the coefficients on the right-hand side of the equation number, and the nonlinear set of state quantities. The above equations are also the basis for the design of specific functions in the simulation. The output equation of the nonlinear prediction model is:

$$\begin{cases} \eta(1) = \dot{x}cos\,\psi - \dot{y}sin\,\psi & \eta(5) = X \\ \eta(2) = \dot{x}sin\,\psi + \dot{y}cos\,\psi & \eta(6) = Y \\ \eta(3) = \dot\phi & \eta(7) = \phi \\ \eta(4) = \dot\psi & \eta(8) = \psi \end{cases}, \tag{68}$$

### 3.2. Constraints

The upper controller based on NMPC requires the construction of constraints, some of which are hardware limitations of the vehicle and some are soft constraints that are used to keep the vehicle in a stable and controllable state. The advantage of the NMPC controller is that it can build nonlinear constraints, making the control results more accurate and improving the system's robustness. Therefore, the following constraints are defined.

$$0 \le v_x \le 160, \tag{69}$$

$$-0.615 \le \dot{v}_x \le 0.615, \tag{70}$$

$$-15 \le \delta \le 15, \tag{71}$$

$$-0.85 \le \dot{\delta} \le 0.85, \tag{72}$$

$$(F_{l,\Omega S})^2 + (F_{c,\Omega S})^2 \le (\mu_{\Omega S}F_{z,\Omega S})^2, \tag{73}$$

The constraints (69) and (70) are vehicle longitudinal speed constraints. (71) and (72) are the equivalent wheel deflection angle constraints.

The vehicle center-of-plane slip angle is defined as:

$$\beta = arctan\left(\frac{v_y}{v_x}\right), \tag{74}$$

where $V_y$ is the lateral velocity of vehicle center-of-plane, $V_x$ is the lateral velocity of vehicle center-of-plane.

Defining center-of-plane slip angle constraints as follows:

$$-5 \leq \beta \leq 5\,, \tag{75}$$

$$-25 \leq \dot{\beta} \leq 25\,, \tag{76}$$

The constraints (75) and (76) are the vehicle center-of-plane slip angle constraints, and the lateral deflection angle is limited to this range, which can make the vehicle move in the state stability region.

Inequality (73) is a tire friction circle constraint; the longitudinal and lateral forces of the tire coupled need to be located within the friction circle of the tire vertical load to prevent tire slip.

*3.3. Cost Function*

The discrete cost function is set as follows:

$$J(\xi(t), U(t)) = \sum_{i=1}^{N_p-1} \|\eta(k+i) - \eta_{ref}(k+i)\|_Q^2 + \sum_{i=1}^{N_c} \left[ \|\Delta u(k+i)\|_R^2 \right], \\ + \|\eta(k+N_p) - \eta_{ref}(k+N_p)\|_P^2 + \rho\varepsilon^2 \tag{77}$$

where $\xi(t)$ is the system state vector trajectory, $\eta(k)$ is the output variables, $\eta_{ref}(k)$ is the desired output variables, $U(t) = [u(k), \cdots, u(k+N_p)]$ is the sequence of control variables. $N_p$ in the time domain, $\Delta u(k+1) = u(k+1) - u(k)$ is the control increment, $N_p$ is the prediction time domain, $N_c$ is the control time domain, $Q$ is the output variables weight, $R$ is the control increment weight, $P$ is the terminal output weight, $\rho$ is the relaxation weight, and $\varepsilon$ is the relaxation factor.

The first term in the cost function is used to make the tracking error between the system output and the desired output as small as possible; the second term is used to penalize the amount of change in the control variables, i.e., the redundant control actions of the system, so that the control variables change smoothly; the third term is used to constrain the terminal output; and the fourth term is a relaxation factor to improve the convergence speed of the operation so that the nonlinear programming can be solved optimally.

*3.4. Solutions*

Based on the above constructed parts, the NMPC control is transformed into solving the following nonlinear programming problem.

$$minJ(\xi(t), U(t)), \tag{78}$$

The Sequential Quadratic Programming (SQP) algorithm is a good solution to the problem [20]. The SQP algorithm consists of four main steps: initialization, quadratic programming solution, line search, and Hessian matrix update, and the optimal control sequence $U^*(t) = [u^*(k), \ldots, u^*(k+N_p)]$ is obtained by solving (65) for each control cycle.

The first element $u^*(k)$ of the optimal control sequence $U^*(t)$ is entered into the system as the actual control variables and executed to the next step. In the new step, the system solves the nonlinear programming problem again according to the new state to obtain the new optimal control sequence, and so on.

## 4. Lower Decoupling-Controller

The research object is a 4WS-4WD vehicle, while the upper controller chose $u = (\delta, a_x)$ as the two-dimensional control variables for the global convexity of the strong coupling nonlinear optimization problem and the solution efficiency. The 4WS-4WD vehicles need eight-dimensional control variables as input variables. The lower decoupling control scheme is constructed to decouple the two-dimensional control variables output by the upper NMPC controller into eight-dimensional control variables that can actually be input

to the 4WS-4WD vehicle. The eight-dimensional control variables are $u = (\delta_{ij}, T_{ij})$, $i \in (f, l)$, $j \in (l, r)$, where $\delta_{ij}$ is wheel angle, $T_{ij}$ is wheel torque. The lower decoupling controller is divided into a four-wheel angle decoupling controller and a four-wheel torque decoupling controller.

*4.1. Four-Wheel Angle Decoupling Control Scheme*

The front-wheel angle decoupling controller is based on Ackerman-steering-geometry, and the rear-wheel angle decoupling controller is based on state-feedback. According to Ackerman's geometric relationship, it can be deduced that the angle relationship between the left and right wheels of the front axle should meet the following formula:

$$cot\,\delta_{fr} - cot\,\delta_{fl} = \frac{2d_r}{l_r + l_f}, \tag{79}$$

And $\delta$ is the comprehensive steering of the left wheel angle $\delta_{fl}$ and right wheel angles $\delta_{fr}$ of the front axle, so it meets the following formula:

$$\delta = \frac{\delta_{fl} + \delta_{fr}}{2}, \tag{80}$$

The rear wheels of four-wheel steer vehicles also have independent steering ability. Considering the real-time angle control of the front wheel, the longitudinal speed, and the yaw rate of the vehicle, a rear wheel angle control scheme based on state feedback is designed as follows:

$$\delta_r = K_1 \delta + K_2(\dot{x})\dot{\psi}, \tag{81}$$

where $K_1$ and $K_2$ are proportional coefficients; $K_1$ is the constant proportional coefficient, $K_1 = -\frac{k_f}{\dot{x}k_r}$; $K_2$ is the proportional coefficient function whose independent variable is the state quantity, $K_2 = \frac{l_f k_f - l_r k_r - m\dot{x}}{\dot{x}k_r}$; $m$ is the vehicle plane, $K_f$ is the lateral stiffness of the front wheel, $K_r$ is the lateral stiffness of the rear wheel.

Meanwhile, based on the Ackerman steering geometry, it can be deduced that the angle relationship between the left and right wheels of the rear axle meets the following formula:

$$cot\,\delta_{rr} - cot\,\delta_{rl} = \frac{2d_f}{l_r + l_f}, \tag{82}$$

Based on the above formula, the four-wheel angles of a vehicle can be uniquely determined under any working condition and state. A four-wheel angle decoupling control scheme is shown in Figure 7.

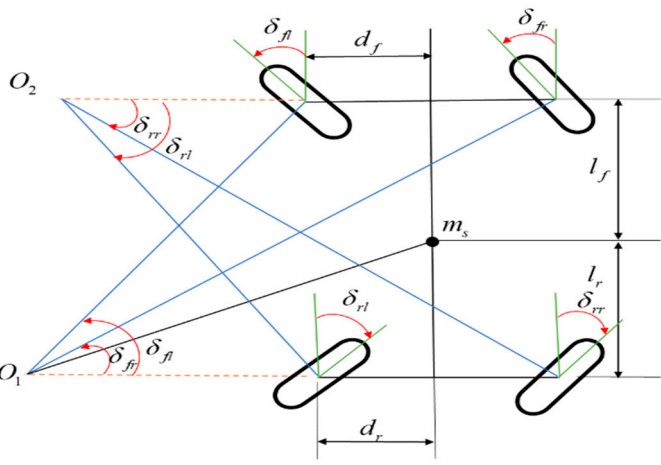

**Figure 7.** Four-wheel angle decoupling control scheme.

*4.2. Four-Wheel Torque Decoupling Control Scheme*

Four-wheel steering four-wheel drive vehicles have four independent drive motors, which can control the driving torque of one wheel, respectively. By controlling the driving or braking torque of the four wheels, the overall acceleration and braking control of the vehicle can be realized. Based on the characteristics of 4WS-4WD vehicles, a multi-objective four-wheel torque decoupling controller is designed. Considering the various objectives of vehicle stability, safety, and tracking accuracy, a unified objective function is established to realize the four-wheel torque decoupling controller.

### 4.2.1. Four-Wheel Torque Control Variables Variation

The variation of four-wheel torque control variables is added to the unified objective function to limit the change of four-wheel torque so vehicles can track their trajectory through minimum control variable adjustment.

The four-wheel torque is as follows:

$$T = \tau_1 T_{fl} + \tau_2 T_{fr} + \tau_3 T_{rl} + \tau_4 T_{rr}, \tag{83}$$

where $\tau_i$ is the weight of four-wheel torque; $T_{ij}(i \in (f, r), j \in (l, r))$ is the wheel torque. The objective function is designed and sorted into quadratic form as follows:

$$
\begin{aligned}
J_1 &= \begin{bmatrix} T_{fl} & T_{fr} & T_{rl} & T_{rr} \end{bmatrix}
\begin{bmatrix} \tau_1 & & & \\ & \tau_2 & & \\ & & \tau_3 & \\ & & & \tau_4 \end{bmatrix}
\begin{bmatrix} T_{fl} \\ T_{fr} \\ T_{rl} \\ T_{rr} \end{bmatrix}, \\
&= u^T H_1 u = \parallel H_1 u \parallel_2^2
\end{aligned}
\tag{84}
$$

### 4.2.2. Upper Control Variable Difference

The lower four-wheel torque is decoupled from the upper control variables, so the difference between them should be added to the unified objective function. The relationship between the lower four-wheel torque and the upper control variables is as follows:

$$
a_x = H_2 u = \begin{bmatrix} \frac{cos\delta_f}{mr_w} & \frac{cos\delta_f}{mr_w} & \frac{cos\delta_{nl}}{mr_w} & \frac{cos\delta_r}{mr_w} \end{bmatrix}
\begin{bmatrix} T_f \\ T_f \\ T_r \\ T_m \end{bmatrix},
\tag{85}
$$

The objective function is designed and sorted into quadratic form as follows:

### 4.2.3. Tire Adhesion Limit

The tire forces generated in tire footprint areas have an upper limit, which is called the tire adhesion limit. The limit resultant forces of a tire under a combined working condition can be obtained by solving the limit of longitudinal and lateral forces in the linear region, respectively, and synthesizing through the "magic formula" model. By extending a series of joint working conditions, the limit resultant tire forces can be expressed by a function family, which is called the "friction circle," which is shown in Figure 8.

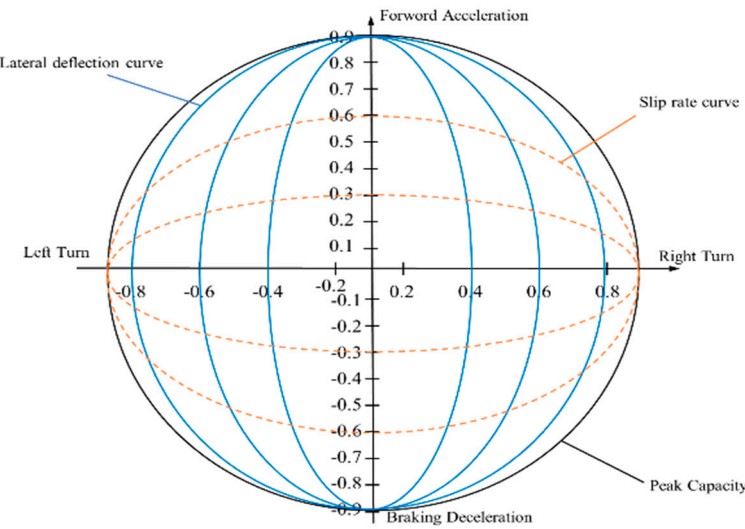

**Figure 8.** Curve of "friction circle" function family.

The function of the "friction circle," which tire longitudinal and lateral forces need to satisfy, is as follows:

$$\left(\frac{F_{xij}}{F_{x\max}}\right)^2 + \left(\frac{F_{yij}}{F_{y\max}}\right)^2 \le 1, (i \in (f,r), j \in (l,r)), \tag{86}$$

Ignoring the influence of lateral forces and taking the limit of the longitudinal force of the tire as the main control object, the objective function is established as follows:

$$
\begin{aligned}
J_3 \quad &= \sum_{i,j} \frac{F_{xij}^2}{(\lambda F_{zij})^2} \\
&= \frac{F_{xfl}^2}{(\lambda F_{zfl})^2} + \frac{F_{xfr}^2}{(\lambda F_{zfr})^2} + \frac{F_{xrl}^2}{(\lambda F_{zrl})^2} + \frac{F_{xrr}^2}{(\lambda F_{zrr})^2} \\
&= \begin{bmatrix} T_{fl} & T_{fr} & T_{rl} & T_{rr} \end{bmatrix} \begin{bmatrix} \frac{1}{\lambda r_w F_{zfl}} & & & \\ & \frac{1}{\lambda r_w F_{zfr}} & & \\ & & \frac{1}{\lambda r_w F_{zrl}} & \\ & & & \frac{1}{\lambda r_w F_{zrr}} \end{bmatrix} \begin{bmatrix} T_{fl} \\ T_{fr} \\ T_{rl} \\ T_{rr} \end{bmatrix} \\
&= u^T H_3 u = \| H_3 u \|_2^2
\end{aligned}
\tag{87}
$$

Based on the above-established objective functions $J_1$ $J_2$ $J_3$, the overall objective function is formed, and the optimization problem is as follows:

$$
\begin{aligned}
minJ \quad &= \omega_1 J_1 + \omega_2 J_2 + \omega_3 J_3 \\
&= \omega_1 \| H_1 u \|_2^2 + \omega_2 \| a_x - H_2 u \|_2^2 + \omega_3 \| H_3 u \|_2^2
\end{aligned}
, \tag{88}
$$

$$\text{s.t } T_{\min} < T_{ij} < T_{\max}, \tag{89}$$

The above optimization problem is a linear quadratic programming problem, which can be solved as the QP (quadratic programming) problem to obtain the lower four-wheel torque control variables.

## 5. Simulation and Analysis

In order to verify the control effect of the hierarchical trajectory tracking controller, a joint Simulink-Carsim simulation experiment is designed in this section to simulate and analyze the trajectory tracking of the vehicle. Among them, Simulink is the simulation

experiment platform, the hierarchical trajectory tracking controller is implemented by Matlab-Function Writing, and the vehicle model of Carsim is called the simulation model.

The simulation experiment platform uses the Driving Scenario Designer in the Matlab toolbox to construct the reference trajectory of the simulated vehicle and imports it into the Simulink simulation platform as the desired output variables $\eta_{ref}$. The C-Class vehicle model of Carsim is selected as the simulation model and connected to Simulink in the form of an S-Function. The NMPC algorithm requires setting specific values for parameters. The parameter settings of NMPC for the hierarchical controller designed in this paper can be seen in Table A1 of Appendix A.

The control variables of the hierarchical trajectory tracking controller can be inputted into Carsim software(Carsim.2020.0) through the Simulink platform. After the control variables are applied to the vehicle model, the Carsim software outputs the driving state of the vehicle to the Simulink platform. The state is transmitted to the hierarchical trajectory tracking controller to complete a cycle of trajectory tracking and feedback correction.

*5.1. Simulation of High-Speed Serpentine Working Conditions*

Figure 9 shows the reference trajectory of the vehicle traveling the serpentine trajectory. The initial speed of the vehicle is 70 km/h and is maintained.

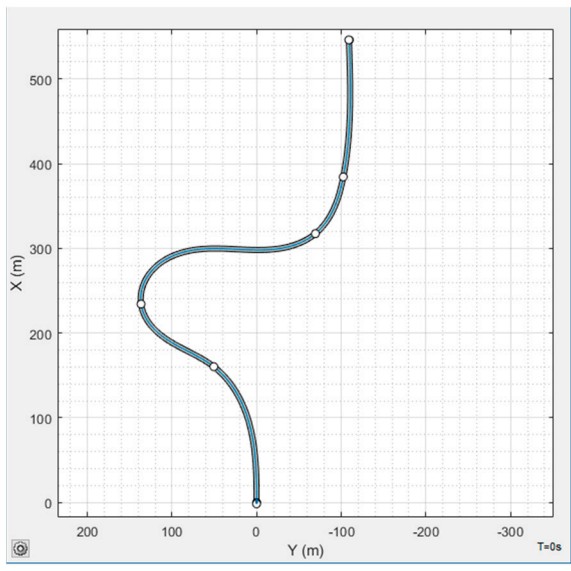

**Figure 9.** High-speed serpentine trajectory.

The simulation results are shown in Figures 10–14. The processed reference trajectory, the simulated trajectory tracking trajectory of the classical MPC model, and the trajectory tracking trajectory of the model in this paper are shown in Figure 10. Figure 10 indicates the trajectory tracking effect of the hierarchical trajectory tracking controller, which shows that the control effect of the hierarchical trajectory tracking controller is good under the high-speed serpentine condition and that the actual trajectory effectively tracks on the desired trajectory. MPC simulation reveals diminished performance at bends. However, our hierarchical trajectory tracking controller efficiently follows the required course at bends, maintaining the lateral distance error within narrow limits when compared with traditional MPC.

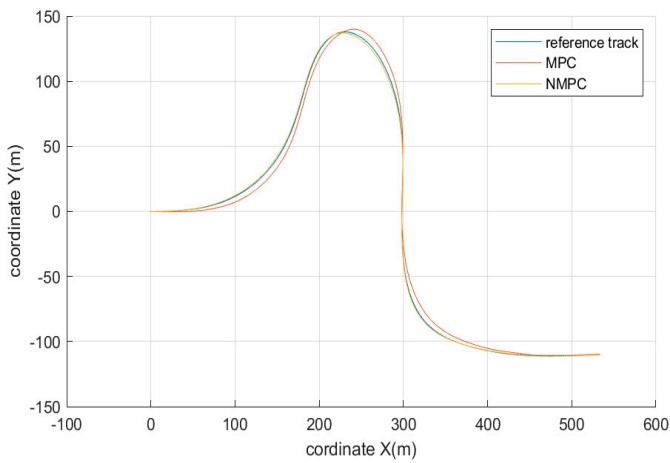

**Figure 10.** Trajectory tracking results of high-speed serpentine.

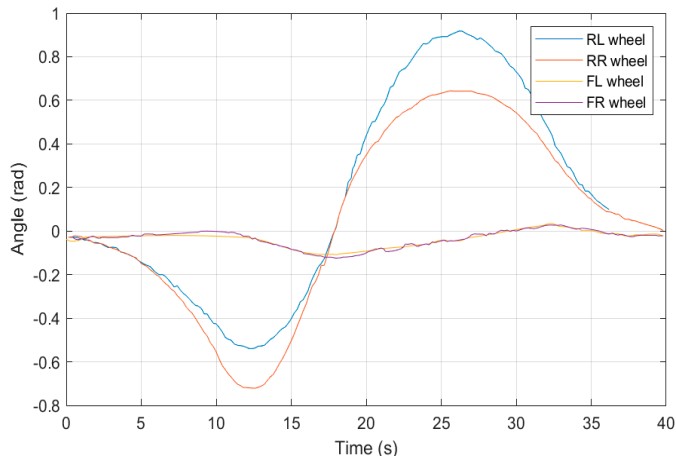

**Figure 11.** Control variable four-wheel tire angle.

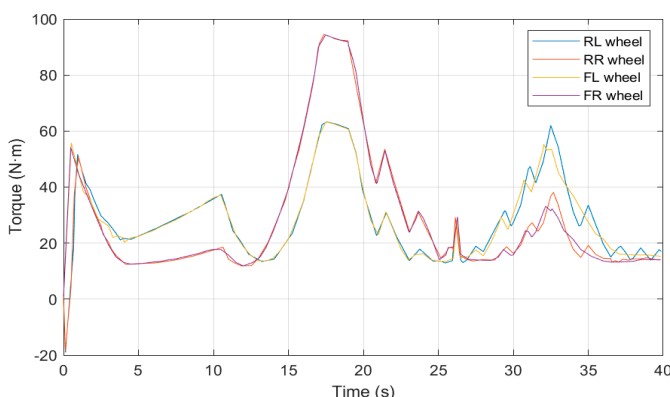

**Figure 12.** Control variable four-wheel torque.

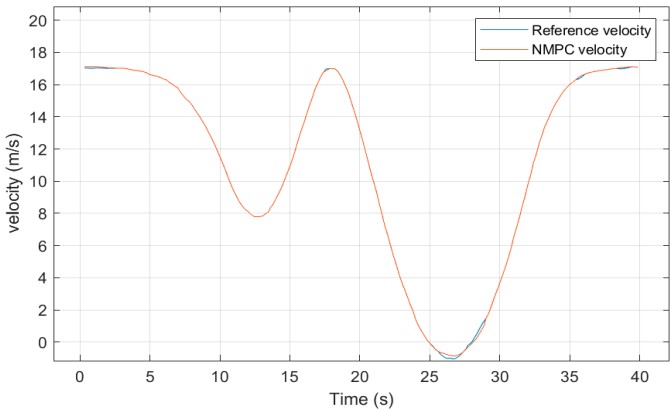

**Figure 13.** Vehicle state variable $\dot{X}$.

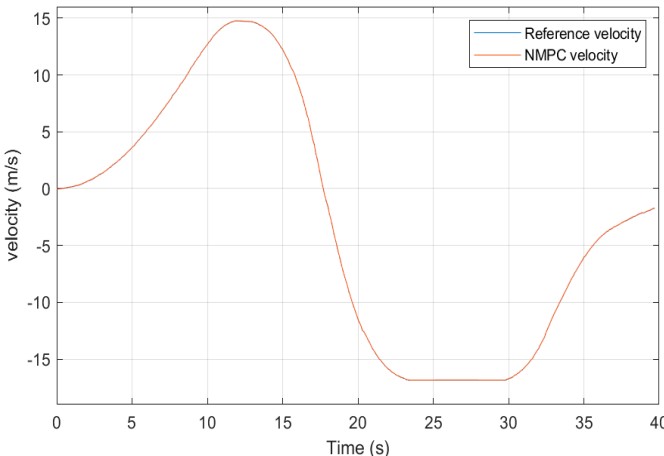

**Figure 14.** Vehicle state variable $\dot{Y}$.

As part of the vehicle motion modeling improvement, especially for the 4WS4WD, the angles and torques of its four wheels need to be studied. Figures 11 and 12 display the change curves for the control variables of four-wheel tire angle and four-wheel torque. Four-wheel drive torque at curved roads with large curvature decreases to improve the stability and trajectory tracking accuracy of the vehicle in curved sections and increases in straight sections to compensate for the time difference between the curvature The four-wheel drive torque is reduced at curved sections to improve the stability and trajectory tracking accuracy of the vehicle in curved sections; it is increased at straight sections to compensate for the time difference between the deceleration at curved sections and the trajectory sequence. Among the four wheel angles, the front wheel angle plays the role of steering, so the control quantity value is small and the response is fast. In four-wheel cornering, the front wheel cornering plays the role of steering, so the control volume value is small and quick response; the rear wheel cornering plays the role of stabilization and feedback to compensate for the vehicle's traverse angular velocity, so in the corners where the traverse angular velocity is large, the rear wheel cornering plays the role of stability. The rear wheel angle is also larger at the corners of the road.

Figures 13–15 show the changes in the motion state of the vehicle. The actual speed $\dot{Y}$ and $\dot{X}$ of the hierarchical trajectory tracking controller in the ground reference system can track the desired speed well, and the initial speed fluctuation can be eliminated quickly, which indicates that the speed control of the controller is robust; the yaw angle $\psi$ of the controller can also track the desired yaw angle $\psi_{ref}$ accurately. The speed tracking of the controller is more accurate, and there is no obvious lag and fluctuation in the yaw angle tracking and speed tracking. $\dot{X}$ fluctuates slightly at the turn, and the speed change trend

is slightly larger than the desired speed, i.e., there is a slight oversteer phenomenon, but it can be compensated and re-tracked on the desired trajectory relatively quickly, which also reflects the overall controller's better robustness.

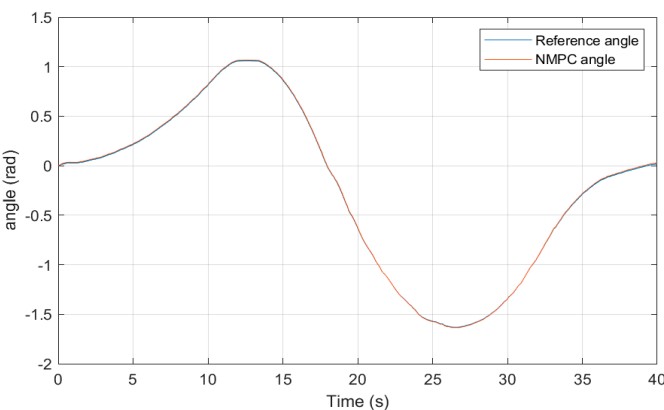

**Figure 15.** Tracking result of yaw angle $\psi$.

### 5.2. Simulation of Double-Shifted Line Working Condition

Simulation conditions are shown in Figure 16. The sample curve is used to draw a double-shifted line vehicle driving trajectory; the initial speed of the vehicle is 60 km/h, and the speed is maintained while traveling.

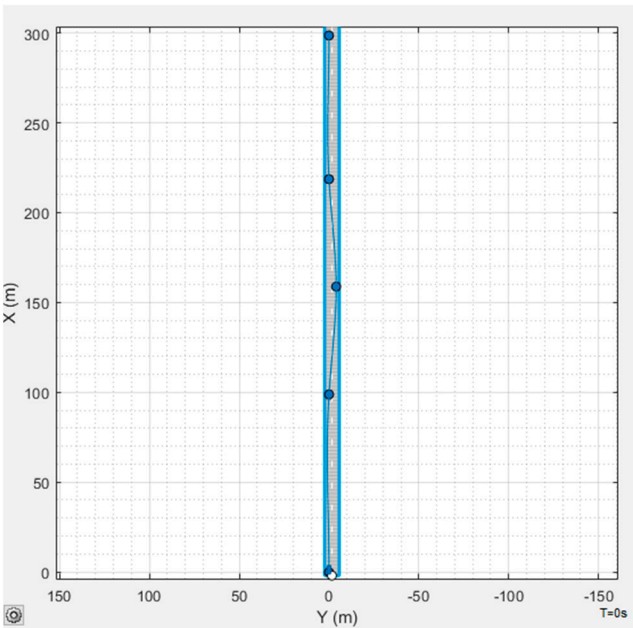

**Figure 16.** Double shift line work trajectory.

The simulation results are shown in Figures 17–21. Figure 17 indicates the trajectory tracking result of the hierarchical trajectory tracking controller, and it can be seen that the control result of the controller is significantly efficient under the double shift line condition, and the lateral position accuracy and yaw angle accuracy of the controller at the bend are good, and the lateral error is controlled within a very small range. The trajectory of the controlled vehicle completely tracks on the desired trajectory and completes the driving maneuver for the double shift line condition with negligible tracking error. This is partly due to the reduced speed of the vehicle; on the other hand, compared with the serpentine condition, the controlled vehicle is in the double-shifted line condition, which has a small turning arc. On the other hand, it shows that there is little room for improvement in

this paper compared to the control effect of the classical MPC in scenarios similar to the double shift line condition. As can be seen in the locally enlarged view of Figure 17b, the improvement of NMPC with respect to MPC is not significant, and in terms of the global trajectory, there is essentially an overlap.

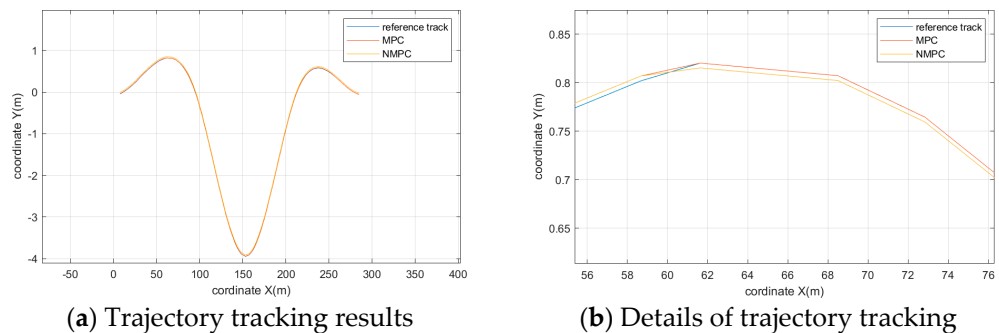

(**a**) Trajectory tracking results         (**b**) Details of trajectory tracking

**Figure 17.** Trajectory tracking results of double-shift line working conditions.

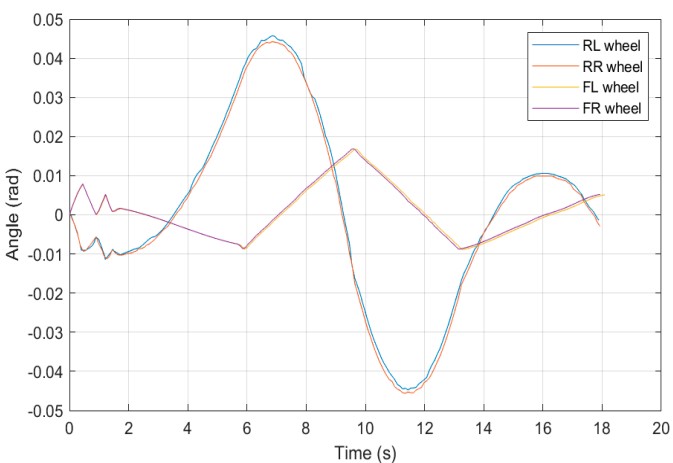

**Figure 18.** Control variable four-wheel tire angle.

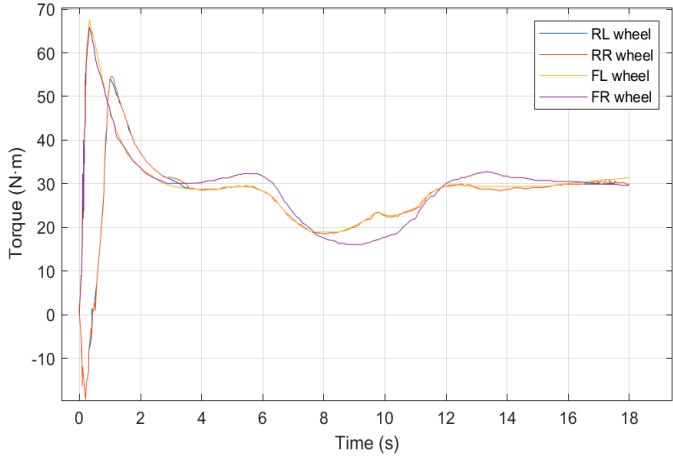

**Figure 19.** Control variable four-wheel torque.

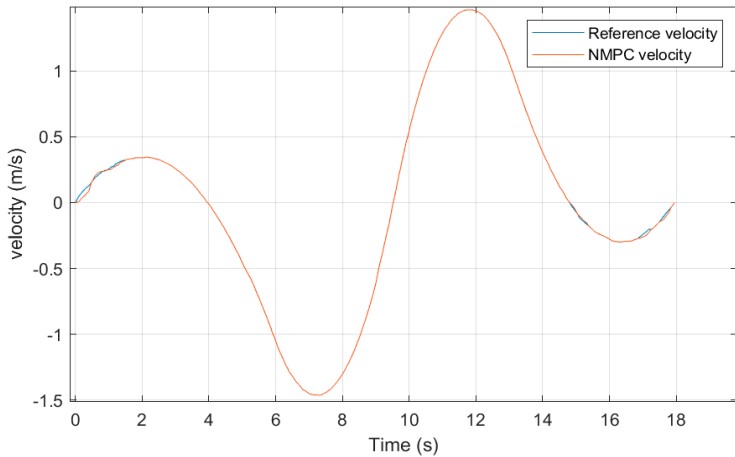

**Figure 20.** Vehicle state variable $\dot{Y}$.

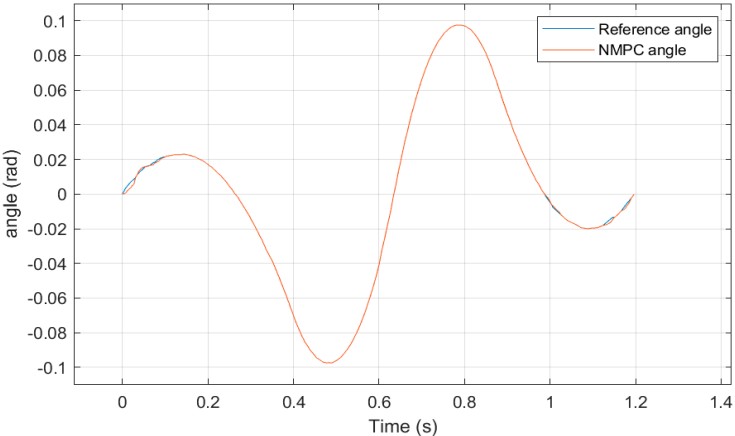

**Figure 21.** Tracking result of yaw angle $\psi$.

As seen in Figures 18 and 19, the control variables of the controller have a substantial change in the four-wheel tire angle to complete the lane change motion of the vehicle. The lower four-wheel torque distribution algorithm assigns that the four-wheel drive motors are larger in the vehicle start-up phase, the drive torque decreases when the vehicle is steering in a double-shift line to complete the steering maneuver, and the driving torque is kept low in the final straight line section to ensure smooth driving of the vehicle. The four-wheel turning angle of the vehicle Between the front and rear axles, the left and right wheels of the front and rear axles are allocated based on Ackermann's formula, and the values of the control quantities are relatively close to each other, and the front axle turning angle changes faster. The front axle angle changes faster in order to respond quickly to the lane-changing action; the rear axle angle is based on state quantity feedback, and the change is smoother to improve the stability of the vehicle.

As seen in Figures 20 and 21, the controller has an accurate tracking effect at the bend during the double shift lane movement, and the errors of both speed and yaw angle are significantly small.

In general, this chapter builds a Matlab/Simulink/Carsim joint simulation platform in order to verify the control effect of the nonlinear trajectory tracking controller based on hierarchical control. Among the trajectory tracking objects are those generated by the desired trajectory generator with Driving Scenario Designer, which corresponds to the reference trajectory generated by the trajectory planning for unmanned vehicles. The nonlinear trajectory tracking control algorithm based on hierarchical control is verified by constructing the simulation tests of the double-shift line condition and the serpentine

condition, and the simulation results show that the algorithm can effectively track the desired trajectory with smaller path tracking errors and higher speed tracking accuracy. Under the working conditions of higher speed and excessive turning, the control effect is greatly improved compared with that of the classical MPC. In straight-line acceleration or turning under smaller working conditions, because the turning nonlinear characteristics are not abrupt, the improvement space is smaller, but it can also meet the vehicle trajectory tracking requirements.

## 6. Conclusions

In this paper, a hierarchical trajectory tracking controller for 4WS-4WD autonomous driving is designed with the following main features.

A three-dimensional attitude change vehicle model is developed, which improves the accuracy of the vehicle model and the ability to capture the nonlinear characteristics of the vehicle by taking into account the effects of attitude change on the longitudinal and lateral control of autonomous driving in the vehicle model. The improved 'magic formula' tire model modifies the time-varying parameters based on the vehicle attitude change geometry and load transfer effects and captures the non-linear characteristics of the tire longitudinal and lateral force outputs. The improvements focus on considering the effects of attitude changes on the longitudinal and lateral control of autonomous driving in the vehicle model.

A two-part hierarchical trajectory tracking controller was designed. The upper trajectory tracking controller is constructed based on the vehicle model, and the lower decoupling controller decouples the upper control variables into eight-dimensional control variables, i.e., four-wheel torque and four-wheel angle.

The Control Strategy based on the NMPC algorithm and 4WS-4WD vehicle characteristics is optimized.

A joint simulation platform based on Carsim and Simulink was established, and simulation experiments were conducted. The results show that, based on the improved vehicle model, the hierarchical trajectory tracking Control Strategy can accurately complete the longitudinal and transverse trajectory tracking control of the vehicle, effectively track the high-speed trajectory under a variety of working conditions, and maintain the tracking accuracy and robustness.

**Author Contributions:** Conceptualization, J.Y. and X.X.; methodology, Q.L.; software, X.X. and K.W.; validation, X.X. and K.W.; formal analysis, X.X.; investigation, X.X.; resources, J.Y.; data curation, Q.L.; writing—original draft preparation, X.X.; writing—review and editing, X.X. and J.Y.; visualization, X.X. and K.W.; supervision, J.Y.; project administration, J.Y.; funding acquisition, J.Y. All authors have read and agreed to the published version of the manuscript.

**Funding:** This research was supported by the Graduate Research Innovation Program Project of Jiangsu Province, China (KYCX22_1059).

**Data Availability Statement:** The data that supports the findings of this study is available from the corresponding author upon reasonable request.

**Acknowledgments:** The authors would like to thank the anonymous reviewers for their valuable suggestions.

**Conflicts of Interest:** On behalf of all authors, the corresponding author states that there are no conflicts of interest.

**Appendix A**

**Table A1.** Parameters and description of the training data files.

| Parameter Name | Parameter | Numeric |
|---|---|---|
| Sampling Period | $T_s$ | 0.01 |
| Prediction Time Domain | $H_p$ | 10 |
| Control Time Domain | $H_c$ | 4 |
| Weighting Matrix | $Q$ | [1,1,0.1,0.2,1,1,0,1] |
| Weight Matrix | $R$ | [0.6,0.6] |
| Weight matrix | $P$ | [0.1,0.1,0,0,0.1,0.1,0,0] |
| Weight Coefficients | $\rho$ | 1000 |
| Weighting Matrix | $H_1$ | [0.3,0.3,0.8,0.8] |
| Weight Coefficients | $\omega_1, \omega_2, \omega_3$ | [0.3 1 0.6] |

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
