# Peer review of "An Optimal Hierarchical Control Strategy for 4WS-4WD Vehicles Using Nonlinear Model Predictive Control"

_machines, doi:10.3390/machines12010084_

Round 1

Reviewer 1 Report

Comments and Suggestions for Authors

The topic of the article is relevant. The Introduction reviews enough new sources, but most of them are discussed rather superficially. The Introduction could be more detailed and at least some of the main sources could be covered in a little more depth.

The Section 2 describes the vehicle model. A very detailed description of the model is provided. The question is, how original is this model and how much is the contribution of the authors themselves to this model? In my opinion, it should be highlighted what is taken as a basis from other sources and what is created by the authors themselves. Section 3 discusses the upper trajectory tracking controller and Section 4 deals with lower decoupling-controller.

In order to verify the control effect of hierarchical trajectory tracking controller, a joint Simulink-Carsim simulation experiment was designed. The results of simulation and their analysis are presented in Section 5. Unfortunately, the results of this simulation are discussed very superficially, more are presented in the figures and not discussed much. The figures show the desired and simulated results only. Lack of discussion of how accurate this model is, what the inputs are, and how they influence the results, and there is little discussion of how the data in Section 5 relates to the material that has preceded it. I think it should be discussed at the end of the article, maybe it is worth making a separate section for the discussion of results. The discussion Section should highlight the contribution of the authors to the problem under consideration, the novelty of the research compared to other recent research in the field, etc. This should also be reflected in the Conclusions.

Author Response

Dear Reviewer.

Thank you very much for your comments and guidance on this paper, here are my revisions:
1 The introduction section has been rewritten with a discussion on model improvement, hierarchical control, and the sources and research gaps of NMPC;
2 The second section provides additional information on the origin and development of MAGIC FORMULA, and some other changes and additions have been made.
3 The discussion section in Chapter 5 has been rewritten to discuss the results of the simulations; and additional introductions have been made in Chapter 3 and elsewhere to better clarify the correspondence; and a separate discussion of the advantages and disadvantages has been made at the end of Chapter 5.
4 The overall revision of this paper is reflected in the end of the file name "editor version" of the word file, you can "revise" mode for review; in addition to the PDF version of the annex and the revised word file backup.

At the end of this letter, wishing you a happy holiday!

Kind regards,

Jiafu Yang;Xuan Xu;Qiongqiong Li;Kang Wang;

Reviewer 2 Report

Comments and Suggestions for Authors

In this article, the authors present a hierarchical design for developing a control system for a vehicle with 4WS-4WD capabilities. The presented modeling is very comprehensive and introduces multiple aspects that realistically capture the dynamic characteristics of the modeled vehicle, including constraints associated with the state variables and control inputs of the modeled system.

However, there are several aspects that, in the opinion of this reviewer, need to be modified:

1.- Due to the proposed algorithm, it is necessary to estimate or indicate in some way the computational load associated with this algorithm. Is it possible to execute it in real-time?

2.- On page 15, equation (68), the article seems to lack content that is not provided. What is the purpose of equation (68) if the nabla element is already defined earlier on page 14?

3.- The presentation of the results and the description of the graphs need to be rewritten and improved to match the level of the rest of the article.

4.- To ensure reproducibility of the results, it would be advisable to publish the code for the performed tests in a public and accessible repository.

Comments on the Quality of English Language

No comments

Author Response

Dear Reviewer.

Thank you very much for your comments and guidance on this paper, here are my revisions and responses:
1 The effect caused by position change and load transfer needs to add corresponding modules in simulink and create FUNCTIONS in programming language, so it is executed in real time in simulation. (You have pointed out a good idea, in this paper, when considering the load, it is only used as a signal optimisation module to correct the tyre model.)
2 Regarding the duplication of definitions at Eq. 68, additional explanations have been made. (It is possible to express it parametrically, the expansion is to illustrate the specific expression of its functions in the simulation).
3 The discussion section in Chapter 5 has been rewritten to discuss the results of the simulation.
4 For reproducibility verification, the main flow of formula derivation and simulation has been further supplemented in the text with Appendix parameter descriptions.
5 The overall revision of this paper is reflected in the PDF file with "editor version" at the end of the file name.

Today is New Year's Day. Wishing you a happy New Year's Day!

Kind regards,

Jiafu Yang;Xuan Xu;Qiongqiong Li;Kang Wang;

Reviewer 3 Report

Comments and Suggestions for Authors

Dear Authors,

The work concerns the modeling of an autonomous vehicle for various control variants. The manuscript is quite extensive and presents new research results, but unfortunately there is too little literature, which is a weak point. In general, the issues discussed in the manuscript and the form of presentation are correct, but there are some issues that need to be clarified and supplemented.

Detailed comments on the manuscript:

1. The abstract should be shortened, in my opinion it is too long.

2. The keywords are correct.

3. In the introductory part you can add more references in the first sentence, I suggest the latest articles:

https://doi.org/10.20858/sjsutst.2021.110.14. https://doi.org/10.3390/en14185778
https://doi.org/10.14669/AM.VOL84.ART2
https://doi.org/10.3390/app12062993
https://doi.org/10.1016/j.trpro.2020.02.031
10.1109/CogMob55547.2022.10117768
https://doi.org/10.20858/sjsutst.2018.100.2.

and

article about 4 × 4 High Mobility Wheeled Vehicle:
https://doi.org/10.3390/en16041938

4. Fig 1., the drawing is of poor quality.

5. The mathematical apparatus is significant in the manuscript.

6. The captions in Fig. 8 could be larger.

7. There are minor editing errors, e.g. line 510, 515, 522, 533.

8. There is no summary of chapter 5, the chapter cannot end with a drawing, we need commentary and discussion with other researchers at the end of this chapter.

9. The conclusions are correct.

10. The literature should be supplemented with the latest articles (20 references are too few).

Thank you

Author Response

Dear reviewer.

Thank you very much for your comments and guidance on this paper, the following are my revisions and responses:
1 The abstract has been rewritten and shortened.
2 The Introduction has been rewritten and discussed on model improvement, hierarchical control, and the sources and research gaps of NMPC. The references you recommended have been added to the first sentence. The references have been reorganised and increased in number.
3 Figure 1 has been redrawn and some other images have been redrawn.
4 The captions in Figure 8 are in mdpi format and could not be made larger. I assume you are talking about the font being too small in the figure, so it has been made larger.
5 The editing error you pointed out has been fixed.
6 The discussion section in Chapter 5 has been rewritten to discuss the results of the simulation.
7 The overall revision of this paper is reflected in the PDF file with "editor version" at the end of the file name.

Today's New Year's Day. Wishing you a happy holiday!

Kind regards,

Jiafu Yang;Xuan Xu;Qiongqiong Li;Kang Wang;

Round 2

Reviewer 1 Report

Comments and Suggestions for Authors

Authors have significantly improved their manuscript.

The quality of some of the Figures in Section 5 could be better, as the writing is hard to read in some places.

Author Response

Dear reviewers

Thank you very much for your comments and guidance on this paper. The errors you pointed out have been corrected. All the images in the Chapter 5 section were redrawn. Thick lines affecting the observation were replaced with thin lines, smoothed curves were restored, and the original data were used for plotting. Some of the pictures with high trajectory overlap have been added with local magnification.

Kind regards.

Jiafu Yang; Xuan Xu; Qiongqiong Li; Kang Wang.

Reviewer 3 Report

Comments and Suggestions for Authors

Dear Authors,

Thank you for your changes to the manuscript and your responses. I accept the corrections, the manuscript is much richer, so I accept it with small corrections. I noticed that reference 7 has already been used earlier, so I suggest replacing it with this reference:
https://doi.org/10.1515/eng-2020-0006

There are still minor editing errors in the work, but I hope that the authors will fix them at a later stage of the publication process.

Thank you

Author Response

Dear reviewers.

Thank you very much for your comments and guidance on this paper. The errors you pointed out have been corrected. Reference 7 has been re-replaced and formatted; the editorial content has been rechecked in its entirety; and the changes are reflected in the PDF file.

Kind regards.

Jiafu Yang; Xuan Xu; Qiongqiong Li; Kang Wang.
